# Safety and Effectiveness of OnabotulinumtoxinA in Patients with Laryngeal Dystonia: Final Report of a 52-Week, Multicenter Postmarketing Surveillance Study

**DOI:** 10.3390/toxins15090553

**Published:** 2023-09-05

**Authors:** Shigeomi Iimura, Yasuyo Nose, Keiko Tabata, Kenji Oda, Yoshiyuki Yamashita, Naohiro Takahashi, Yoshiaki Kawano

**Affiliations:** 1VEO Specialty Care, Value Evidence Outcome, Japan Medical and Development, GlaxoSmithKline K.K., Tokyo 107-0052, Japan; shigeomi.2.iimura@gsk.com (S.I.); yasuyo.x.nose@gsk.com (Y.N.); 2PMS Data Management, Value Evidence Outcome, Japan Medical and Development, GlaxoSmithKline K.K., Tokyo 107-0052, Japan; keiko.x.tabata@gsk.com; 3Real World Data Analytics, Value Evidence Outcome, Japan Medical and Development, GlaxoSmithKline K.K., Tokyo 107-0052, Japan; kenji.x.oda@gsk.com; 4Specialty Medical Affairs, Japan Medical and Development, GlaxoSmithKline K.K., Tokyo 107-0052, Japan; yoshiyuki.2.yamashita@gsk.com; 5PMS, Value Evidence Outcome, Japan Medical and Development, GlaxoSmithKline K.K., Tokyo 107-0052, Japan; naohiro.2.takahashi@gsk.com

**Keywords:** onabotulinumtoxinA, laryngeal dystonia, postmarketing surveillance study

## Abstract

This postmarketing surveillance study was conducted to evaluate the safety and effectiveness of onabotulinumtoxinA in Japanese patients with laryngeal dystonia (LD). Patients receiving onabotulinumtoxinA for the first time were enrolled and observed for up to 12 months following the first injection. Safety assessment included adverse drug reactions (ADRs), and effectiveness assessments included the Voice Handicap Index-10 (VHI-10) and physician’s global assessment (PGA). ADRs were observed in 48 (5.8%) of 834 patients in the safety analysis set, including dysphonia in 43 (5.2%) patients and dysphagia in 7 (0.8%) patients. The change in total VHI-10 score (mean) in 790 patients included in the effectiveness analysis set showed that improvement in adductor LD peaked at 2 months after the first injection, while patients with abductor or mixed LD showed a gradual attenuation of effect 2–4 weeks post-injection. The change in total VHI-10 score in subsequent injections was generally similar to that following the first injection. The overall effectiveness rate according to the PGA was 93.4% (738/790 patients). The results demonstrate that onabotulinumtoxinA is a well-tolerated and effective treatment for LD in real-world clinical practice.

## 1. Introduction

Laryngeal dystonia (LD), also known as spasmodic dysphonia, is a speech disorder with no structural abnormality in the phonatory organs, with the symptoms caused by involuntary and intermittent spasm of the internal laryngeal muscles during phonation [1,2]. The pathophysiology of this disease is focal dystonia, classified into three types: adductor, abductor, and mixed. The most common type is adductor LD. In adductor LD, expiratory airflow is blocked during phonation as the vocal folds are strongly involuntarily and intermittently adducted, resulting in interrupted voice and lack of smoothness [3], sometimes also producing a squeezing of the vocal folds. The abductor type presents with an intermittent “breathy” voice, falsetto voice, and aphonia due to the involuntary opening of the glottis during phonation [4]. Both adductor and abductor LD result in the inability to carry out conversation smoothly in work and daily life, causing a significant hindrance to one’s social life.

The prevalence of LD has been reported as 0.7/100,000 population in Europe [5]. In a 2015 nationwide epidemiological study in Japan, more than half (59.0%) of patients were relatively young (in their 20s and 30s), with a mean age at onset of 30.9 years and a male-to-female ratio of 1:4.1; the prevalence in Japan was estimated to be 3.5–7.0/100,000 population [6]. There is no curative treatment for the disease, and conservative treatments include vocal training (voice therapy) and oral medicine administration to remove the tension of the larynx during phonation; however, evidence on the effectiveness of these therapeutic options is scarce. Surgical treatment options for adductor LD include resection of the thyroarytenoid muscle (an internal laryngeal muscle) and Type II thyroplasty, which reduces excessive closure of the vocal folds during phonation [7]. However, thyroarytenoid myectomy causes hoarseness after surgery due to vocal fold atrophy or scarring, and Type II thyroplasty requires anterior cervical skin incision. In addition, evidence regarding long-term effectiveness of those treatments is not robust.

The most common treatment for LD worldwide is local onabotulinumtoxinA injection therapy to the internal laryngeal muscles. The involuntary adduction and abduction movements of the vocal folds are suppressed by injecting onabotulinumtoxinA primarily into the thyroarytenoid muscle for adductor LD and into the posterior cricoarytenoid muscle for abductor LD. OnabotulinumtoxinA injection can be performed percutaneously through the anterior neck in a short period of time; thus, there is no need for hospitalization. It has been reported that the effectiveness rate of adductor LD is more than 90%, while the effectiveness rate of abductor LD is slightly less [8,9]. The American Academy of Otolaryngology Head and Neck Surgery Clinical Practice Guideline for hoarseness (dysphonia) also recommends local onabotulinumtoxinA injection therapy for LD [10].

In Japan, a physician-initiated domestic phase II/III trial (BTX-SD-01) was conducted [3], and onabotulinumtoxinA was approved for the indication of LD in 2018 based on the results. OnabotulinumtoxinA is the only type A botulinum toxin preparation approved for LD in Japan. This postmarketing surveillance (PMS) study is a condition for approval by the Ministry of Health, Labour and Welfare (MHLW) of Japan, monitoring the real-world safety and effectiveness of onabotulinumtoxinA for LD for up to a 12-month period. Here, we describe the results of this PMS study in 834 patients with LD who received onabotulinumtoxinA in a real-world setting in Japan.

## 2. Results

### 2.1. Patient Disposition

In this study, 1579 patients with LD from 63 sites were enrolled. Among them, case records of 929 patients who initiated onabotulinumtoxinA treatment by 31 December 2018, were collected, and data from their case report forms (CRFs) were finalized on 28 December 2021. The number of patients with finalized CRFs in the safety analysis set was 834, excluding a total of 95 patients (94 patients lost to follow-up after the initial administration and 1 patient who had already received onabotulinumtoxinA at another hospital). In the 95 patients who were excluded from the safety analysis set, no adverse drug reactions (ADRs) were reported. The effectiveness analysis set consisted of 790 patients, excluding 44 patients who were not evaluable (Figure 1).

### 2.2. Patient Characteristics

The characteristics of the 834 patients included in the safety analysis set are shown in Table 1. There were more females than males (83.1% vs. 16.9%), and the mean age was 41.6 years. Adductor LD was found in 96.2% of patients, abductor LD in 2.8%, and mixed LD in 1.1%. The mean age at onset was 32.0 years, and 37.2% reported a disease duration of ≥10 years. The initial dose (mean ± standard deviation) was 2.59 ± 1.30 units for the thyroarytenoid muscle and 5.40 ± 1.18 units for the posterior cricoarytenoid muscle.

In the 9 patients with mixed LD, 6 patients received the injection into only one side of the thyroarytenoid muscle and the other 3 patients received the injection into only one side of the posterior cricoarytenoid muscle at the initial administration. Multiple injections were performed in 6 patients with a maximum of five injections. In 4 patients, the injected muscle was alternated for repeated injections, switching between thyroarytenoid and posterior cricoarytenoid muscles or between the left and right posterior cricoarytenoid muscles. In the remaining 2 patients, the injections were repeated to the ipsilateral thyroarytenoid muscle. The other 3 patients received a single injection. In most cases, the dose was 2.5–5.0 units in the posterior cricoarytenoid muscle or 1.0–2.5 units in the thyroarytenoid muscle for unilateral administration; in some cases, bilateral administration of 2.5 units to each thyroarytenoid muscle (total of 5.0 units) was performed. The characteristics of the 790 patients in the effectiveness analysis set were similar to those in the safety analysis set.

### 2.3. Safety

#### 2.3.1. Incidence of Adverse Drug Reactions

In the 834 patients included in the safety analysis set, 48 patients reported ADRs, with an incidence rate of 5.8% (Table 2).

The ADRs reported included “dysphonia” in 43 patients, “dysphagia” in 7 patients, “choking” in 6 patients, “aspiration” in 3 patients, and “pneumonia aspiration” in 2 patients. Among those, all cases of “choking”, “aspiration”, and “pneumonia aspiration” were classified as serious ADRs. All three cases of “aspiration” and two cases of “pneumonia aspiration” reported as serious ADRs were in patients with adductor LD. Those events were transient, and all patients recovered. There were no clear associations between the proportion of patients with ADRs and the onabotulinumtoxinA dose or injected muscle. Only 1 patient received bilateral injections of 5.0 units into the posterior cricoarytenoid muscle (the highest single dose in this study), and the patient had no ADRs.

The incidence of ADRs after repeated administration is shown in Table 3. The proportions of patients who experienced an ADR after their first, second, third, and fourth administrations of onabotulinumtoxinA were 3.8% (32/834 patients), 2.6% (16/614 patients), 1.4% (6/427 patients), and 1.3% (3/226 patients), respectively; no ADRs were reported after the fifth administration (0/52 patients).

#### 2.3.2. Incidence of Adverse Drug Reactions of Special Interest

Regarding the occurrence of adverse drug reactions of special interest (ADRSIs) in the safety analysis set, only ADRs related to dysphagia and possible distant spread of toxin (PDSOT) were reported; no ADRs related to other safety specifications were reported. ADRs related to dysphagia were observed in 11 patients (incidence of 1.3% with 95% confidence interval (CI) of 0.7–2.3) (Table 4). They included “dysphagia” in 7 patients, “aspiration” in 3 patients, and “pneumonia aspiration” in 2 patients. Of these, all cases of “aspiration” and “pneumonia aspiration” were classified as serious ADRs. All patients recovered.

ADRs related to PDSOT were observed in 47 patients, with an incidence of 5.6% (95% CI: 4.2–7.4) (Table 4). These included 43 patients with “dysphonia” and 11 patients with ADRs related to dysphagia (mentioned above).

### 2.4. Effectiveness

Among the 790 patients included in the effectiveness analysis set, there were 762 patients with adductor LD (Table 1). Among these patients, total Voice Handicap Index-10 (VHI-10) score (measured value and change) over time was analyzed (Figure 2A). The mean (±95% CI) total VHI-10 score at the first dose of onabotulinumtoxinA improved from 24.8 ± 0.7 at baseline to 13.3 ± 1.0 at 2 months post-injection, after which the score gradually increased. Trends in total VHI-10 scores for the second to fifth injections were generally similar, with the greatest improvement from baseline to 1–2 months after injection, followed by a gradual increase in scores.

Among the 790 patients included in the effectiveness analysis set, 22 patients had abductor LD (Table 1). Among these patients, the change in total VHI-10 score (measured value) was determined (Figure 2B). Mean (±95% CI) total VHI-10 scores at the first dose of onabotulinumtoxinA improved from 28.1 ± 6.2 at baseline to 18.9 ± 8.5 at 2 weeks post-injection. Trends in total VHI-10 scores between the second and fourth injections were roughly similar, with the greatest improvement from baseline to 2 weeks–1 month after injection; however, due to the small sample size, subsequent scores did not show a clear trend. For the fifth dose, only 1 patient was included; however, the baseline score at the fifth dose was low, and an increase in the score was observed after onabotulinumtoxinA administration.

Of the 790 patients included in the effectiveness analysis set, there were 6 patients with mixed LD, and the change in total VHI-10 score (measured value and change) was analyzed. The mean total VHI-10 score at the first dose of onabotulinumtoxinA was 24.4 ± 9.5 at baseline. No improvement was observed in total VHI-10 score, although the number of cases was limited. Similarly, for the second to fifth doses, the sample size was small and the data were highly variable.

Physician’s global assessment (PGA) of effectiveness was also evaluated in this study. Investigators comprehensively assessed the effectiveness at 12 months after the starting day of onabotulinumtoxinA administration or at the time of discontinuation/completion of observation, according to the course of subjective symptoms during the observation period from the initial administration day and the course of clinical symptoms (VHI-10), and either effectiveness, ineffectiveness, or inability to determine was established. In the effectiveness analysis set, the proportion of patients whose treatment was judged to be effective (effectiveness rate) was 93.4% (738/790 patients), and the ineffectiveness rate was 6.6% (52/790 patients). The investigators were unable to determine treatment effectiveness in 44 patients, and these patients were excluded from the effectiveness analysis set (Figure 1). According to disease type, the effectiveness rate was 93.8% (715/762 patients) in adductor LD, 81.8% (18/22 patients) in abductor LD, and 83.3% (5/6 patients) in mixed LD. Overall, there was no clear association between onabotulinumtoxinA dosage and the proportion of responders.

## 3. Discussion

At the time this study was planned, onabotulinumtoxinA for LD was approved in Australia and some Latin American countries, but not in the United States, Europe, or Asia; thus, many individuals with LD were not able to benefit from onabotulinumtoxinA [3]. In Japan, onabotulinumtoxinA was approved for treatment of LD in 2018 based on the results of the BTX-SD-01 study. However, the BTX-SD-01 study enrolled only 24 patients; therefore, there was not sufficient evidence regarding safety and effectiveness in a large Japanese cohort. To date, the present study is one of the largest prospective observational studies of LD patients treated with onabotulinumtoxinA.

Of the 834 patients included in the safety analysis set, 83.1% were female, 96.2% had adductor LD, and the mean age at onset was 32.0 years, which was similar to previous epidemiological studies in Japan [6]. Although studies have been performed in other countries that reported a higher percentage of females than males among LD patients [8,11], the present study included an even higher proportion of females. This may be explained by differences between this study and studies from other countries in the study methods and in the genes related to the onset of LD, as discussed by Hyodo et al. [6]. It has also been found that Japanese women are more sensitive to voice disorders than men [6].

In this study, ADRs were reported in 48 of 834 patients (5.8%) in the safety analysis set. The reported ADRs included 43 cases of “dysphonia”, 7 cases of “dysphagia”, 6 cases of “choking”, 3 cases of “aspiration”, and 2 cases of “pneumonia aspiration”. In the BTX-SD-01 study, the most-frequent adverse events (AEs) were voice disorder and swallowing disorder [3]. Previous studies have also reported mild and mild-to-moderate paralytic dysphonia (breathiness) in 35% and 23% of patients, respectively [8,11]. Since dysphonia as an ADR was assessed according to the investigator’s subjective judgement, based on the patient’s voice condition at the interview, it may have been under-reported in this study. These ADRs are temporary and usually improve within a few days to weeks. Dysphonia has been reported under AE names such as hoarseness or voice disorder, which are thought to correspond to breathiness. It is likely that these events are the result of the toxin’s effect on the injected muscles. There were no clinically significant differences in patient characteristics (age, sex, disease duration) between those with and without dysphagia. Interestingly, dysphagia was not observed in abductor LD patients in the BTX-SD-01 study [3]. Dysphagia has been reported to occur after onabotulinumtoxinA treatment; however, its incidence was lower in the present study than previously reported [8,11]. Dysphagia may be related to the diffusion of onabotulinumtoxinA into the inferior constrictor muscles [11].

Serious ADRs were not observed in the BTX-SD-01 study [3]; however, 11 serious ADRs of “choking”, “aspiration”, and “pneumonia aspiration” were reported in this study. All cases of choking were reported by investigators as non-serious events caused by cough reflex due to small amounts of stray fluid in the trachea. However, the only Medical Dictionary for Regulatory Activities (MedDRA) preferred term (PT) that corresponds to those events is “choking”, which is regarded as serious (defined as a medically significant event by EU regulatory authority [12]). All cases of “aspiration” were also classified as serious ADRs for the same reason. Four out of five cases of “aspiration” and “pneumonia aspiration” were in patients over 50 years old. In the BTX-SD-01 study, the mean age was 38.5 years, and no serious adverse reactions occurred. Further studies are warranted to clarify if age is a risk factor for serious aspiration and pneumonia aspiration events. All of the serious ADRs reported in this study were reversible.

The incidence of adverse reactions following the first, second, third, and fourth administration of onabotulinumtoxinA was 3.8% (32/834 patients), 2.6% (16/614 patients), 1.4% (6/427 patients), and 1.3% (3/226 patients), respectively. No adverse reactions were reported after the fifth administration. Despite differences in the number of patients between onabotulinumtoxinA doses, there was no tendency for increased incidence or severity of adverse reactions or the occurrence of new adverse reactions with repeated administration of onabotulinumtoxinA. In the BTX-SD-01 study, the incidence of adverse reactions did not differ substantially during the first dose versus subsequent doses. In addition, as there were no serious safety concerns associated with repeated administration of onabotulinumtoxinA in the reported studies [8,13,14] with detailed descriptions of safety in other countries, the safety risk was not considered to be increased by repeated administration of onabotulinumtoxinA.

Of the events related to the safety specifications in this study, only those related to dysphagia and PDSOT were reported; no other events were reported. All reported events related to dysphagia were those with a higher incidence in the BTX-SD-01 study [3]. The only ADRs reported as events related to PDSOT were dysphagia and dysphonia, which are considered local events due to toxin diffusion and the site of administration of onabotulinumtoxinA, respectively.

In adductor LD patients in the BTX-SD-01 study, the change from baseline in total VHI-10 score (mean ± standard deviation) 4 weeks after the first dose of onabotulinumtoxinA was −8.3 ± 10.05. In this study, the first injection of onabotulinumtoxinA had a change of −10.7 ± 11.3 (95% CI: −11.74 to −9.73) at 1 month from baseline after the first dose. Although a direct comparison cannot be made due to differences in the patient characteristics, methods of the studies, and sample size, this result suggests that the effectiveness of the first injection in patients with the adductor type shown in the BTX-SD-01 study is reproducible in the real-world clinical setting.

In abductor LD patients in the present study, the change in total VHI-10 score from baseline to 1 month after injection was −8.2 ± 15.1, demonstrating improvement in total VHI-10 score. In the BTX-SD-01 study, the change in total VHI-10 score from baseline to 4 weeks after the first dose of onabotulinumtoxinA was 0.5 ± 0.71; improvement in total VHI-10 score was not observed. However, a direct comparison is difficult due to the limited number of patients; in the BTX-SD-01 study, there were only two cases of abductor LD in the full analysis set.

In the present study, the mean change from baseline in total VHI-10 score was −8.2 ± 15.1 in the abductor type compared with −10.7 ± 11.3 in the adductor type. OnabotulinumtoxinA is considered somewhat less effective in abductor LD than adductor LD [15], and outcomes are not always satisfactory compared with those achieved in patients with adductor LD. One possible reason for this is the relative difficulty in administering onabotulinumtoxinA to the posterior cricoarytenoid muscle. This is thought to be a limitation that results in less satisfactory results in patients with abductor LD compared with adductor LD [16]. Another potential reason for the low efficacy of onabotulinumtoxinA in abductor LD is that bilateral administration of onabotulinumtoxinA to the posterior cricoarytenoid muscle, which is performed worldwide [17], is prohibited in Japan due to safety concerns. However, it is reported that a change of 6 points in total VHI-10 score may represent a minimal important difference (MID), which is the “smallest difference in score in the domain of interest which patients perceive as beneficial” [18]. In this study, the change in total VHI-10 score was similar to or more than the MID in both adductor and abductor types, suggesting that onabotulinumtoxinA injection may also be useful for abductor LD patients.

The VHI-10 is a patient-subjective assessment of activity and participation, not body functions. It does not include an objective assessment by a healthcare worker and is a different approach to objective measurements such as aerodynamic inspection or acoustic analysis [19]. Using the VHI-10 alone, without combining it with these objective assessments, may make it difficult to fully assess the essential characteristics of the voice problem and the effectiveness of treatment. For these reasons, while the VHI-10 is useful as a single measure, given its characteristics, it may have limitations when used as an outcome measure on its own.

In this study, a single patient received a fifth dose of onabotulinumtoxinA (for abductor LD), and the “5th injection” line in Figure 2B represents that one patient’s change in total VHI-10 score. The treatment plan for this patient, per the investigator’s decision, involved administering the fifth dose of onabotulinumtoxinA 2 weeks after the fourth dose, which is shorter than the 12-week minimum interval recommended in the Japanese package insert. As a result, the patient may have had a lower-than-expected baseline total VHI-10 score at the time of the fifth dose, which could have influenced the resulting observed change in total VHI-10 score.

The effectiveness rate in the PGA during the follow-up was 93.4% (738/790 patients). PGA was an investigator-assessed “yes” or “no” judgement of treatment effectiveness, based on whether the patient’s symptoms and VHI-10 score had improved. PGA was limited by the potential for bias because it is a subjective assessment by the investigator rather than the result of independent assessments by several reviewers. Although this was not performed in other studies, this result supports the overall benefit of onabotulinumtoxinA for LD. Also of note, effectiveness was observed in 5 out of 6 patients with mixed LD in this analysis. Although the number of patients was limited, this is the first paper demonstrating the effectiveness of onabotulinumtoxinA in this disease type. It should be noted that the results of this study, including those regarding the other types of LD, are specific to onabotulinumtoxinA and should not be extrapolated to any other botulinum toxin, as dosing units of these products are not interchangeable.

## 4. Conclusions

In this study, no new safety issues were identified and the effectiveness reported in previous studies was reproduced in the real-world practice of onabotulinumtoxinA treatment in Japanese LD patients. This indicates that onabotulinumtoxinA injection is a beneficial option for the treatment of LD.

## 5. Materials and Methods

This study was conducted in Japan as a prospective observational study in accordance with the Ministry of Health, Labour and Welfare (MHLW) Ordinance No. 171 dated December 20, 2004 (Good Postmarketing Surveillance Practice) concerning the standards for the implementation of postmarketing surveillance and studies of drugs. Therefore, approval by the ethics committee at each site and informed consent were not required. No personally identifiable information was collected.

All patients who received onabotulinumtoxinA for LD after approval in Japan on 25 May 2018, were eligible for this study, and those who started onabotulinumtoxinA treatment by 31 December 2018, were enrolled based on approval by the local regulatory agency. The patient observation period was up to 12 months from the date of the first injection of onabotulinumtoxinA.

### 5.1. Patient Characteristics

Sex, age, type of LD, age at onset of LD, medical history (renal dysfunction, hepatic dysfunction, other diseases), the presence of concomitant diseases (renal dysfunction, hepatic dysfunction, other comorbidities), and prior therapies were investigated.

### 5.2. OnabotulinumtoxinA Treatment

The date of administration, injected muscle, and dosage of onabotulinumtoxinA were investigated. Any change in the injected muscle and dosage in subsequent administrations and the reason for the change were also reported.

### 5.3. Safety Evaluation

For all AEs that occurred after onabotulinumtoxinA administration, the event name, date of onset, outcome, seriousness, and relationship with onabotulinumtoxinA were investigated. Serious ADRs were defined as (1) death from an AE, (2) symptoms that could lead to death (life-threatening), (3) symptoms that required hospitalization or prolongation of hospitalization for treatment, (4) symptoms that were permanent or resulted in significant disability/insufficiency, (5) symptoms that resulted in congenital abnormalities or defects, or (6) events or reactions that were considered to be otherwise medically significant. Those AEs judged by the investigator or the sponsor as events for which a relationship with onabotulinumtoxinA could not be ruled out were considered ADRs. ADRs reported by the investigator were coded by the PT using MedDRA/J Version 24.1. In order to clarify the usage of the MedDRA/J Version 24.1 terms in this paper, they are described with double quotation marks. For example, investigator-reported ADRs such as hoarseness, breathy hoarseness, or voice disorder were coded as “dysphonia” by MedDRA/J Version 24.1.

In addition, this study investigated the occurrence of the following events as safety specifications as a part of the risk management plan of onabotulinumtoxinA for LD approved by the local regulatory agency: (1) hypersensitivity reactions, (2) administration to patients with neuromuscular disorders, (3) production of neutralizing antibodies, (4) dysphagia, (5) possible distant spread of toxin, (6) seizure, (7) interactions with muscle relaxants, and (8) interactions with other botulinum toxin products administered concurrently or with an interval of several months. ADRs related to these safety specifications were defined as ADRSIs in this paper. ADRSIs were defined by the MedDRA/J Version 24.1 terms as shown in Appendix A.

### 5.4. Effectiveness Evaluation

#### 5.4.1. Voice Handicap Index-10

Patients completed a voice-related questionnaire for VHI-10 assessment at each injection visit. VHI-10 assessment points were immediately before the first dose, at Week 2, and at Months 1, 2, 3, 6, 9, and 12. At the time of reinjection (after the second dose), the assessment points were immediately before administration, Week 2, and Months 1, 2, 3, 6, and 9 after administration. When the observation was discontinued within 12 months from the initial administration date, VHI-10 was also evaluated at that time. The longest follow-up time was 12 months after the first dose of onabotulinumtoxinA.

#### 5.4.2. Physician’s Global Assessment

The investigator comprehensively assessed the effectiveness at 12 months after the initial administration date or at the time of discontinuation/completion of observation. Treatment was judged to be either effective or ineffective based on the course of subjective symptoms and the VHI-10 during the observation period. If the assessment of effectiveness could not be determined for any reason, it was considered “undeterminable” by investigators. The effectiveness rate of the PGA was calculated using the number of effective cases divided by the total number of effective and ineffective cases.

### 5.5. Statistical Analysis

Safety was assessed using the MedDRA PTs to calculate the number of patients with ADRs and the incidence rate. The 95% CI for the incidence was calculated using the Clopper–Pearson method.

The 95% CIs for the total VHI-10 score of effectiveness were calculated using *t*-distributions. SAS Ver.9.3J (SAS Institute Inc., Cary, NC, USA) was used for statistical analysis.

## Figures and Tables

**Figure 1 toxins-15-00553-f001:**
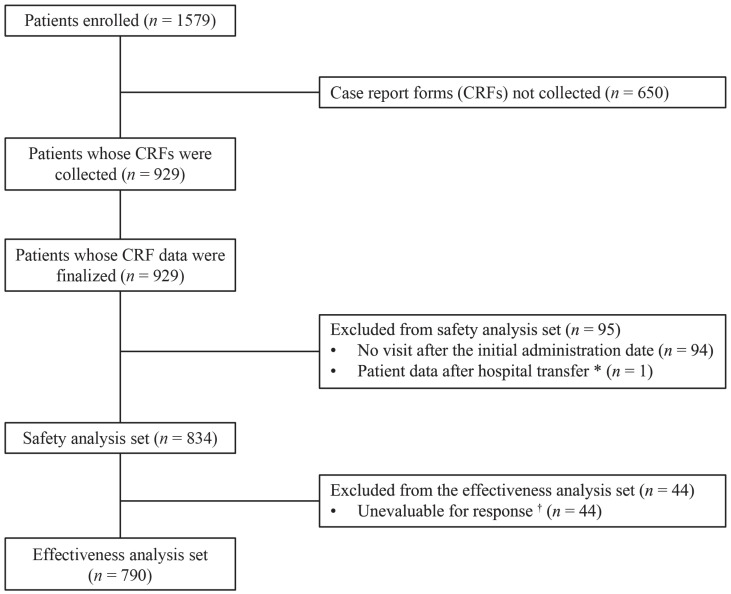
Patient disposition. * Duplicate cases due to hospital transfer, for which CRF data before and after hospital transfer were integrated and data after hospital transfer were excluded. ^†^ Unevaluable cases for response due to reasons such as lost to follow-up. Abbreviations: CRF, case report form.

**Figure 2 toxins-15-00553-f002:**
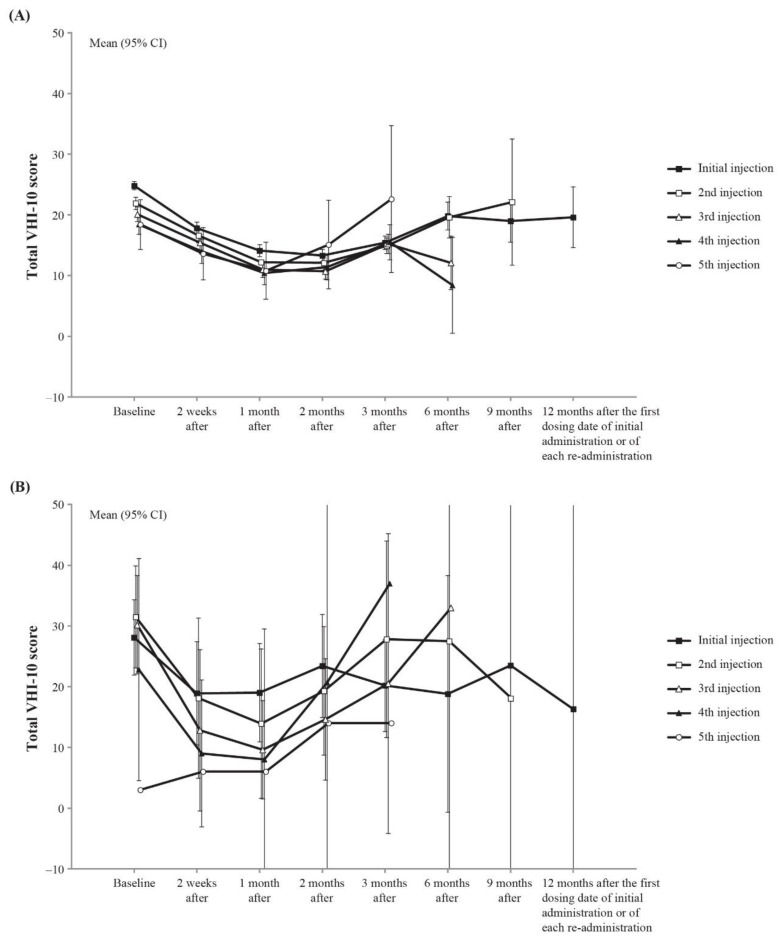
Total VHI-10 score time courses. (**A**) Adductor LD and (**B**) abductor LD. Patients in the effectiveness analysis set, whose scores at baseline and after onabotulinumtoxinA administration were available. Abbreviations: CI, confidence interval; LD, laryngeal dystonia; VHI-10, Voice Handicap Index-10.

**Table 1 toxins-15-00553-t001:** Patient characteristics (safety analysis set and effectiveness analysis set).

Characteristic	Safety Analysis Set	Effectiveness Analysis Set
Total patients	834 (100.0)	790 (100.0)
Sex		
Male	141 (16.9)	129 (16.3)
Female	693 (83.1)	661 (83.7)
Age, years		
Mean ± SD	41.6 ± 14.8	41.8 ± 14.7
Median (min–max)	40.0 (14–87)	40.0 (14–87)
<15 years	1 (0.1)	1 (0.1)
≥15 to <65 years	759 (91.0)	719 (91.0)
≥65 years	74 (8.9)	70 (8.9)
Disease type (at the time of initial onabotulinumtoxinA administration)		
Adductor LD	802 (96.2)	762 (96.5)
Abductor LD	23 (2.8)	22 (2.8)
Mixed LD	9 (1.1)	6 (0.8)
Injected muscle		
Thyroarytenoid muscle	807 (96.8)	764 (96.7)
Posterior cricoarytenoid muscle	21 (2.5)	20 (2.5)
Mixed type	6 (0.7)	6 (0.8)
Age at onset of LD, years		
Mean ± SD	32.0 ± 14.7	32.0 ± 14.6
Median (min–max)	28.0 (9–83)	28.0 (9–83)
<30 years	412 (49.4)	389 (49.2)
≥30 years	368 (44.1)	350 (44.3)
Unknown	54 (6.5)	51 (6.5)
Disease duration, years		
<2 years	119 (14.3)	108 (13.7)
≥2 to <4 years	120 (14.4)	109 (13.8)
≥4 to <6 years	94 (11.3)	89 (11.3)
≥6 to <8 years	87 (10.4)	82 (10.4)
≥8 to <10 years	50 (6.0)	47 (5.9)
≥10 years	310 (37.2)	304 (38.5)
Unknown	54 (6.5)	51 (6.5)
Medical history	41 (4.9)	34 (4.3)
Renal impairment	2 (0.2)	2 (0.3)
Hepatic impairment	1 (0.1)	1 (0.1)
Other diseases	39 (4.7)	32 (4.1)
Comorbidities	28 (3.4)	28 (3.5)
Renal impairment	2 (0.2)	2 (0.3)
Hepatic impairment	0	0
Other	28 (3.4)	28 (3.5)
Prior medications	266 (31.9)	263 (33.3)
Prior therapies	241 (28.9)	213 (27.0)
Surgical therapy	25 (3.0)	22 (2.8)
Type II thyroplasty	24 (2.9)	21 (2.7)
Thyroarytenoidectomy	1 (0.1)	1 (0.1)
Voice therapy	221 (26.5)	195 (24.7)
Other	1 (0.1)	1 (0.1)
Dose (at the time of initial onabotulinumtoxinA administrtion), units *		
Thyroarytenoid muscle	810 (97.1)	767 (97.1)
Mean ± SD	2.59 ± 1.30	2.60 ± 1.33
Median (min–max)	2.50 (0.00625–10.0)	2.50 (0.00625–10.0)
Unilateral injection		
Mean ± SD	2.47 ± 0.99	2.47 ± 1.01
<2.5	130 (15.6)	127 (16.1)
2.5	475 (57.0)	439 (55.6)
>2.5	76 (9.1)	76 (9.6)
Bilateral injections		
Mean ± SD	3.25 ± 2.22	3.28 ± 2.25
<5.0	87 (10.4)	83 (10.5)
5.0	30 (3.6)	30 (3.8)
>5.0	12 (1.4)	12 (1.5)
Posterior cricoarytenoid muscle	25 (3.0)	24 (3.0)
Mean ± SD	5.40 ± 1.18	5.42 ± 1.20
Median (min–max)	5.00 (5.0–10.0)	5.00 (5.0–10.0)
Unilateral injection		
Mean ± SD	5.21 ± 0.71	5.22 ± 0.72
<5.0	0	0
5.0	22 (2.6)	21 (2.7)
>5.0	2 (0.2)	2 (0.3)
Bilateral injections		
Mean ± SD	10.00 ± 0	10.00 ± 0
<10.0	0	0
10.0	1 (0.1)	1 (0.1)
>10.0	0	0

Data are presented as *n* (%) unless otherwise specified. * Duplicates included. Abbreviations: LD, laryngeal dystonia; SD, standard deviation.

**Table 2 toxins-15-00553-t002:** Incidence of ADRs (safety analysis set).

Safety Analysis Set, *n*	834
No. of Patients with ADRs (%)	48 (5.8)
Type of ADRs	No. of patients with ADRs (%)
Infections and infestations	2 (0.2)
Pneumonia aspiration	2 (0.2)
Respiratory, thoracic, and mediastinal disorders	44 (5.3)
Aspiration	3 (0.4)
Choking	6 (0.7)
Dysphonia	43 (5.2)
Gastrointestinal disorders	7 (0.8)
Dysphagia	7 (0.8)

MedDRA/J Version (24.1). Abbreviations: ADR, adverse drug reaction; MedDRA, Medical Dictionary for Regulatory Activities.

**Table 3 toxins-15-00553-t003:** Incidence of ADRs during repeated administration (safety analysis set).

	Overall *		Initial Administration		2nd Administration		3rd Administration		4th Administration		5th Administration	
	Total	Serious	Total	Serious	Total	Serious	Total	Serious	Total	Serious	Total	Serious
Number of patients investigated	834		834		614		427		226		52	
Patients with ADRs	48 (5.8)	11 (1.3)	32 (3.8)	9 (1.1)	16 (2.6)	0	6 (1.4)	2 (0.5)	3 (1.3)	0	0	0
Type of ADRs												
Respiratory, thoracic, and mediastinal disorders	44 (5.3)	9 (1.1)	29 (3.5)	8 (1.0)	15 (2.4)	0	4 (0.9)	1 (0.2)	2 (0.9)	0	0	0
Dysphonia	43 (5.2)	0	28 (3.4)	0	15 (2.4)	0	3 (0.7)	0	2 (0.9)	0	0	0
Choking	6 (0.7)	6 (0.7)	5 (0.6)	5 (0.6)	0	0	0	0	0	0	0	0
Aspiration	3 (0.4)	3 (0.4)	3 (0.4)	3 (0.4)	0	0	1 (0.2)	1 (0.2)	0	0	0	0
Gastrointestinal disorders	7 (0.8)	0	3 (0.4)	0	1 (0.2)	0	1 (0.2)	0	1 (0.4)	0	0	0
Dysphagia	7 (0.8)	0	3 (0.4)	0	1 (0.2)	0	1 (0.2)	0	1 (0.4)	0	0	0
Infections and infestations	2 (0.2)	2 (0.2)	1 (0.1)	1 (0.1)	0	0	1 (0.2)	1 (0.2)	0	0	0	0
Pneumonia aspiration	2 (0.2)	2 (0.2)	1 (0.1)	1 (0.1)	0	0	1 (0.2)	1 (0.2)	0	0	0	0

MedDRA/J Version (24.1). * Including three cases where the onset date of the event was unknown. Data are presented as *n* (%). Abbreviations: ADR, adverse drug reaction; MedDRA, Medical Dictionary for Regulatory Activities.

**Table 4 toxins-15-00553-t004:** Incidence of ADRs related to “dysphagia” and “possible distant spread of toxin” as ADRSIs (safety analysis set).

		Overall	Serious Cases
ADRs related to “dysphagia”	Number of patients investigated	834	834
	Patients with ADRs	11	5
	Type of ADRs		
	Gastrointestinal disorders	7 (0.8)	0
	Dysphagia	7 (0.8)	0
	Respiratory, thoracic, and mediastinal disorders	3 (0.4)	3 (0.4)
	Aspiration	3 (0.4)	3 (0.4)
	Infections and infestations	2 (0.2)	2 (0.2)
	Pneumonia aspiration	2 (0.2)	2 (0.2)
ADRs related to “possible distant spread of toxin”	Number of patients investigated	834	834
	Patients with ADRs	47	5
	Type of ADRs		
	Respiratory, thoracic, and mediastinal disorders	43 (5.2)	3 (0.4)
	Dysphonia	43 (5.2)	0
	Aspiration	3 (0.4)	3 (0.4)
	Gastrointestinal disorders	7 (0.8)	0
	Dysphagia	7 (0.8)	0
	Infections and infestations	2 (0.2)	2 (0.2)
	Pneumonia aspiration	2 (0.2)	2 (0.2)

MedDRA/J Version (24.1). Data are presented as *n* (%). The number of patients with ADRs was tabulated in the following order of priority: fatal > recovered/resolved with sequelae > not recovered/not resolved > recovering/resolving > recovered/resolved > unknown. If multiple events classified into the same system organ class or as the same PT occurred in the same patient, they were tabulated by system organ class or PT in the following order of priority: fatal > recovered/resolved with sequelae > not recovered/not resolved > recovering/resolving > recovered/resolved > unknown. All cases were recovered/resolved with the exception of 1 patient who had non-serious “dysphonia” that was deemed recovering/resolving. Abbreviations: ADR, adverse drug reaction; ADRSI, adverse drug reactions of special interest; MedDRA, Medical Dictionary for Regulatory Activities; PT, preferred term.

## Data Availability

The data presented in this study are available upon request from the corresponding author. The data are not publicly available, and a reasonable restriction on data availability must be implemented because informed consent was not obtained due to the above reason.

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
