# Peer review of "Safety and Effectiveness of OnabotulinumtoxinA in Patients with Laryngeal Dystonia: Final Report of a 52-Week, Multicenter Postmarketing Surveillance Study"

_toxins, 2023, doi:10.3390/toxins15090553_

Round 1

Reviewer 1 Report

This is a large study, comprehensive, well devised and whose conclusions are an excellent contribution to our literature

Author Response

We thank the reviewer for this kind feedback.

Reviewer 2 Report

This is an important paper showing demonstrating safety and efficacy of botulinum toxin for laryngeal dystonia (formerly known as spasmodic dysphonia) and the authors should be congratulated for performing this study which further supports this as the gold standard treatment for LD. I have some suggestions and comments which may improve the paper from a readers perspective.

The accepted nomenclature has changed and the term laryngeal dystonia (LD) should be used rather than spasmodic dysphonia. For historical purposes the term spasmodic dysphonia can be included / defined.

Line 44 - please provide references for surgical resection paragraph

VHI-10 is not necessarily the best outcome measure for spasmodic dysphonia but is acceptable. The authors should acknowledge its limitations in the context of the systematic review done by Rumbach et al on outcome measures in SD

It would be useful to know the mean dose and SD for unilateral vs bilateral TA injections in addition to / instead of just the overall mean

Please reference the PGA. It is unclear what instrument this is and how it was scored. Is this a validated assessment or just the physician’s assessment of whether there was improvement? Was it carried out by the treating doctor or independently? If the former then acknowledgement of potential bias should occur. Please define effectiveness rate and how many patients found the treatment ineffective or inability to assess?

Dysphonia as an ADR is reported at a very low rate of 5.2% The authors acknowledge that paralytic dysphonia (an expected ADR) reported in other papers occurs in up to 1/3 of cases. The difference needs to be explained better. Perhaps the definition of "dysphonia" as an ADR should be discussed. How was it assessed? Objective or subjective? It appears to be “investigator reported” but the meaning of this is unclear. Did the investigator assess the voice perceptually? Did the investigator ask the patient whether their voice was better or worse? There may be reporting bias if the assessment was carried out by the treating physician Other papers have looked at longitudinal outcomes at higher resolution and found dysphonia as reported by patients was maximal in the first 2-3 weeks after injection. The disparity and reasons for this should be discussed in detail and the limitations of this assessment should be acknowledged.

Figure 2B - How do the authors explain the VHI-10 score increasing over 3 months in the group receiving their fifth dose of BTX?

Did the authors look at / find any correlation between improvement in VHI and dose of botox?

Did the authors look at whether there is a relationship between dose / location of botox given and adverse reactions? – I ask this especially regarding the case where PCA muscles were injected 5U bilaterally

Author Response

All page and line numbers refer to the version of the manuscript without tracked changes shown.

Reviewer 3 Report

It's an interesting piece of work. The results are well elaborated and
summarized. I hardly see any gaps. 2 to 3 relevant papers should be added.
Furthermore, the differentiation from other botulinum toxins should be
presented
briefly.  

largely error-free

Author Response

(The authors gave the same response as above.)
